| Open Peer Review | Environmental Microbiology | Methods and Protocols

# Cultivating efficiency: high-throughput growth analysis of anaerobic bacteria in compact microplate readers

Oona L. O. Snoeyenbos-West,[1] Christina R. Guerrero,[1] Makaela Valencia,[1] Paul Carini[1,2,3]

**ABSTRACT**    Anaerobic microbes play crucial roles in environmental processes, industry, and human health. Traditional methods for monitoring the growth of anaerobes, including plate counts or subsampling broth cultures for optical density measurements, are time and resource-intensive. The advent of microplate readers revolutionized bacterial growth studies by enabling high-throughput and real-time monitoring of microbial growth kinetics. Yet, their use in anaerobic microbiology has remained limited. Here, we present a workflow for using small-footprint microplate readers and the Growthcurver R package to analyze the kinetic growth metrics of anaerobic bacteria. We benchmarked the small-footprint Cerillo Stratus microplate reader against a BioTek Synergy HTX microplate reader in aerobic conditions using *Escherichia coli* DSM 28618 cultures. The growth rates and carrying capacities obtained from the two readers were statistically indistinguishable. However, the area under the logistic curve was significantly higher in cultures monitored by the Stratus reader. We used the Stratus to quantify the growth responses of anaerobically grown *E. coli* and *Clostridium bolteae* DSM 29485 to different doses of the toxin sodium arsenite. The growth of *E. coli* and *C. bolteae* was sensitive to arsenite doses of 1.3 µM and 0.4 µM, respectively. Complete inhibition of growth was achieved at 38 µM arsenite for *C. bolteae* and 338 µM in *E. coli*. These results show that the Stratus performs similarly to a leading brand of microplate reader and can be reliably used in anaerobic conditions. We discuss the advantages of the small format microplate readers and our experiences with the Stratus.

**IMPORTANCE**  We present a workflow that facilitates the production and analysis of growth curves for anaerobic microbes using small-footprint microplate readers and an R script. This workflow is a cost and space-effective solution to most high-throughput solutions for collecting growth data from anaerobic microbes. This technology can be used for applications where high throughput would advance discovery, including microbial isolation, bioprospecting, co-culturing, host-microbe interactions, and drug/toxin-microbial interactions.

**KEYWORDS**  anaerobic bacteria, microbial growth, high throughput, compact microplate readers, growth response, arsenite

Measuring microbial growth is crucial for understanding the behavior and physiology of single-celled organisms (1–3). Optical density (OD) measurements are commonly used to monitor microbial growth. The collection of OD data has become more efficient with the advent of automated microplate readers and software packages (4–6). Microplate readers allow many samples to be analyzed simultaneously, providing reliable and efficient measurements of growth rates and other kinetic data. This has significantly improved the accuracy and speed of high-throughput data collection and analysis in microbiology research. For example, microplate readers have been used to study the growth responses to antibiotics or environmental toxins (7–10), the

Address correspondence to Oona L. O. Snoeyenbos-West, oonawest@arizona.edu, or Paul Carini, paulcarini@arizona.edu.

The authors declare no conflict of interest.

See the funding table on p. 9.

chronological life span in yeast (11), metabolic footprinting in *Pseudomonas* (12), and fitness effects in long-term evolution experiments (6), among other applications.

Relative to aerobic microbes, tracking the growth dynamics of anaerobic microbes adds a layer of complexity due to the need for culture manipulations and incubations to be conducted without exposure to oxygen. Researchers typically culture anaerobic microbes in Hungate or Balch-type culture tubes with thick rubber septa (13). Unfortunately, there are no automated tools currently available to measure optical density from cultures incubated in these types of tubes (14), meaning researchers must manually pierce the rubber septa with a needle to extract a small volume of culture to be read by a spectrophotometer. This process is labor-intensive, low throughput, and relies heavily on the efficiency of the researcher conducting the experiments.

Several approaches have been used to circumvent the low-throughput nature of anaerobe cultivation. One approach excluded oxygen by sealing lids onto microtiter plates and flushing the headspace with $N_2$ to form anaerobic conditions (15). A second approach enzymatically removed oxygen from the culture medium ahead of growth analysis (16). These two approaches attempt to limit oxygen exposure while the plate is incubated outside of an anaerobic chamber in oxygenated conditions—an approach that may not work well for all microbes. In contrast, the environment of the microplate reader itself can be modified by relocating it into an anaerobic chamber (17–20). However, many microplate readers are too large to fit through anaerobic chamber airlocks and require connections to an external computer for data collection. Anaerobic chambers are also typically restrictive, providing just enough room for researchers to carry out necessary manipulations and incubations. Additionally, microplate readers are expensive instruments that are financially inaccessible to many laboratories. Because of the high cost, they are often purchased as shared laboratory equipment or are placed in core facilities, where they need to be accessible for a wide variety of applications. Thus, the substantial physical size, the common need for connection to a computer, and the cost of many microplate readers have been obstacles to their widespread adoption in anaerobic chambers (21)

Here, we present a high-throughput cultivation and growth analysis pipeline based on inexpensive small-footprint microplate readers to overcome some of the challenges associated with microbial growth analysis in anaerobic conditions (Fig. 1). These small-format microplate readers—together with a small-footprint shaker—facilitate the collection of $OD_{600}$ measurements from cultures growing in an anaerobic chamber. We show that the mean growth rates and carrying capacities derived from modeling $OD_{600}$ measurements of aerobic bacteria collected with the small-footprint reader did not vary significantly from those rates collected and analyzed with a state-of-the-art popular brand of microplate reader. We used the small-footprint microplate reader to quantify the effect of arsenite exposure on the anaerobic growth of two bacterial strains belonging to the mouse intestinal bacterial collection (22). Our approach is translatable to high-throughput growth studies on anaerobic microbes from any environment.

## RESULTS AND DISCUSSION

We sought to develop a cost-effective high-throughput strategy to measure and model the growth of anaerobic bacteria (Fig. 1). This workflow is based on the Cerillo Stratus microplate reader, which captures $OD_{600}$ and temperature measurements at user-defined time points and stores the data in comma separated value (.csv) or text (.txt) files on a MicroSD card. The output data contain measurement timestamps, time point temperature, and $OD_{600}$ readings for each well of a 96-well microplate. We paired the Stratus microplate reader with a Grant Instruments Microplate Shaker PMS-1000i small-footprint orbital shaker. This shaker was chosen because it fits inside a COY Model 2000, Forced Air Incubator for Vinyl Chambers that are commonly sold with COY anaerobic chambers. We modeled microbial growth using custom scripts and Growthcurver in R (23).

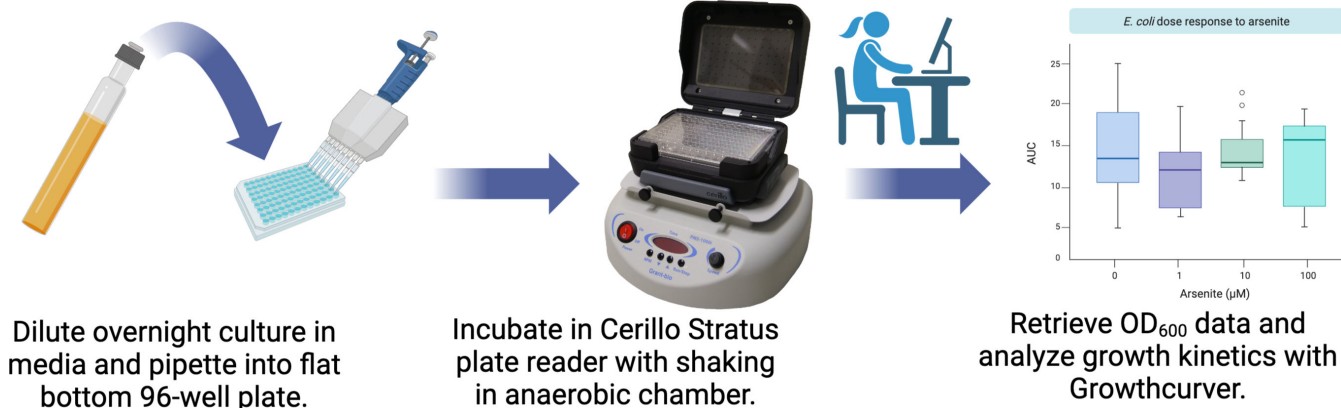

Dilute overnight culture in media and pipette into flat bottom 96-well plate.

Incubate in Cerillo Stratus plate reader with shaking in anaerobic chamber.

Retrieve OD$_{600}$ data and analyze growth kinetics with Growthcurver.

**FIG 1** Conceptual diagram of small-footprint plate reader workflow. Figure constructed with Biorender.com.

We benchmarked the efficacy of the Stratus to obtain accurate growth curve data by comparing the growth rates obtained from the Stratus microplate reader to those obtained from a BioTek Synergy HTX that is commonly used to monitor microbial growth. To do this, we grew aerobic overnight cultures of *Escherichia coli* DSM 28618 and used them to inoculate two 96-well microtiter plates containing lysogeny broth (LB) media. We inoculated 64 wells and left 32 wells uninoculated. The plates were covered with Breathe-Easy Sealing Membranes to facilitate gas and water vapor exchange and placed in either the Synergy HTX or the Stratus reader. The plate in the Synergy HTX microplate reader was incubated at 37°C with continuous shaking at 180 rpm using Synergy HTX's internal temperature control and shaking functionality. For the Stratus reader, the microplate was placed inside the Stratus, and the reader was attached to the PMS-1000i microplate shaker with an adapter supplied by Cerillo. The Stratus was continuously shaken at 180 rpm inside a 37°C incubator (Fig. 1). Both microplate readers were operating outside of an anaerobic chamber in aerobic conditions. The OD$_{600}$ was measured every 3 min overnight (~18–20 h). We downloaded the OD$_{600}$ data from both microplate readers and modeled the resulting kinetic parameters with the Growthcurver R package (23).

Growth rates and carrying capacities from the Stratus and Synergy HTX microplate readers did not differ significantly, but the area under the curve (AUC) displayed significant differences across the two readers (Fig. 2). The mean carrying capacities (*k* in Growthcurver) obtained from the two readers were statistically indistinguishable (Fig. 2a; Mann-Whitney *U* test; $P = 0.088$). The mean carrying capacity obtained from the Stratus was 0.51 ± 0.02 [mean ± standard error of the mean (SEM); $n = 64$] compared to 0.47 ± 0.01 (mean ± SEM; $n = 64$) for the Synergy HTX (Fig. 2a). Similarly, the mean growth rates (*r* in Growthcurver) across the two devices did not differ significantly (Fig. 2b; Mann-Whitney *U* test; $P = 0.094$). The average growth rate of *E. coli* cultures incubated in the Stratus microplate reader was 1.24 ± 0.06 h$^{-1}$ (mean ± SEM; $n = 64$) compared to 1.28 ± 0.02 h$^{-1}$ (mean ± SEM; $n = 64$) for cultures grown in the Synergy HTX. Growthcurver calculates a metric that integrates growth information from the logistic parameters derived from *r* and *k* called AUC (*auc_l* in Growthcurver). The AUC values derived from the Stratus (10.2 ± 0.36 mean ±SEM; $n = 64$) were significantly higher than those obtained from the Synergy HTX (7.65 ± 0.21 mean ± SEM; $n = 64$) (Fig. 2c; Mann-Whitney *U* test; $P = 1.14e{-}14$).

The growth data collected from the Stratus plate were bimodally distributed for all three growth parameters (Fig. 2). Inspection of the growth curves showed a growth curve variant that was characterized by a higher carrying capacity and a slower growth rate resulting in a higher AUC (Fig. S1 and S2). This growth variant was identified in both the Synergy HTX and the Stratus-incubated microtiter plates (Fig. S1 and S2, respectively). However, the proportion of wells displaying this variant was higher in the Stratus (33% of

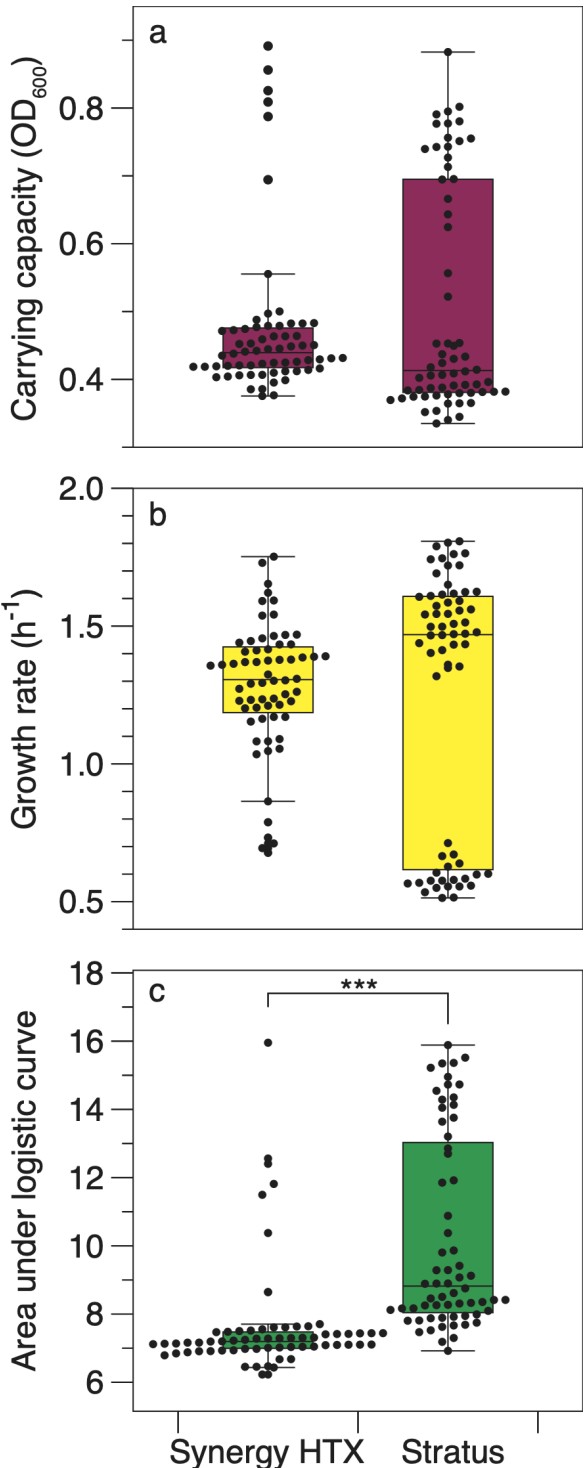

**FIG 2** Stratus plate readers generated mean growth rates and maximum carrying capacities indistinguishable from the Synergy HTX. Points are the kinetic parameters calculated from microbial growth in each inoculated well, including (a) carrying capacity, (b) growth rate, and (c) area under the logistic curve. All values were calculated from $OD_{600}$ measurements by the respective microplate reader using Growthcurver. Each point represents a measurement from a single replicate well. Box plots illustrate the interquartile range ± 1.5× interquartile range. The horizontal line in each box plot is the median. Outliers (>1.5× interquartile range) are shown as points. The bracket with asterisks in (c) denotes a significant difference (Mann-Whitney $U$ test; $P$ value ≤0.001).

wells) than in the Synergy HTX (8% of wells). Previous work in *E. coli* showed growth rate bimodality was dependent on the initial density of cells in each well (24). Yet, in our experiments, both plates were inoculated from the same culture suspension, making significant differences in initial cell densities an unlikely explanation. We speculate that mixing differences between the two shakers—either in actual shaking rate or the orbit diameter—may explain the observed differences. Subtle differences in shaking may have affected the oxygenation of the cultures, leading to the observed growth differences. Wells exhibiting larger AUC are not restricted to certain areas of the plate (e.g., around the edges or in the center; blue boxes in Fig. S1 and S2).

Growth curves collected from the two readers closely resemble each other (Fig. 3). We overlayed the mean $OD_{600} \pm$ SEM ($n = 64$) on a linear plot (Fig. 3a) to illustrate that there is slight variability in the growth curves obtained from the two readers. This variability is most apparent in stationary phase. The Stratus recorded slightly higher maximal $OD_{600}$ readings in stationary phase, and the associated standard error was greater. It is unclear if the differences in the measurement error arose from differences in how the plate readers function, differences in the experimental incubation conditions, or biological reasons. Nonetheless, when these plots are converted to semilogarithmic plots virtually all differences in lag phase, exponential growth kinetics, and stationary phase are negligible (Fig. 3b).

After confirming the efficacy of the Stratus microplate reader in aerobic conditions, we used it in anaerobic conditions. We set up an experiment to quantify how the AUC of *E. coli* DSM 28618 and *Clostridium bolteae* DSM 29485 responded to different doses of arsenite when growing anaerobically. AUC has been used previously to quantify the degree of inhibition in microbial cultures (25, 26). We initiated overnight cultures of *E. coli* or *C. bolteae* from freezer stocks and used the culture to anaerobically inoculate 200 µL cultures in polystyrene microwell plates containing the appropriate growth medium. We added arsenite at eight doses: 0, 0.4, 1.3, 3.8, 38.5, 116, 335, or 1,039 µM with eight replicates per dose. The plates were covered with Breathe-Easy Sealing Membranes to allow gas exchange and incubated with shaking at 180 rpm at 37°C overnight in Stratus microplate readers inside a COY anaerobic chamber with a 95%:5% $N_2$:$H_2$ atmosphere.

The mean AUC for anaerobically grown *C. bolteae* and *E. coli* was significantly variable across the range of concentrations applied (Fig. 4; Kruskal-Wallis *P* value $1.43 \times 10^{-9}$ and $3.48 \times 10^{-8}$, respectively). *C. bolteae* grown in the absence of arsenite had a mean AUC of $14.9 \pm 3.04$ (mean ± standard deviation). AUC was reduced at arsenite doses of 0.4–3.8 µM, relative to no arsenite controls, but not significantly (Fig. 4a; Dunn's Bonferroni-corrected *P* value ≥0.05). We defined complete growth inhibition as treatments with modeled AUC that were statistically indistinguishable from AUC calculated from the uninoculated wells. Complete inhibition of *C. bolteae* was observed at arsenite concentrations greater than 38 µM (Fig. 4a; Dunn's Bonferroni-corrected *P* value ≥0.05 relative to uninoculated blanks). In contrast, *E. coli* displayed a mean AUC of $3.33 \pm 0.25$, and 0.4 µM arsenite had no significant effect on the AUC (Fig. 4b; Dunn's Bonferroni-corrected *P* value >0.05). Arsenite doses of 1.3–116 µM reduced growth, albeit not significantly compared to no arsenite treatments (Fig. 4b; Dunn's Bonferroni-corrected *P* value ≥0.05). Arsenite doses of 335 and 1,039 µM significantly inhibited growth relative to unamended cultures but were significantly higher than rates modeled from the uninoculated wells (Fig. 4b; Dunn's Bonferroni-corrected *P* value ≤0.05). These results are consistent with previous studies on the inhibitory effect of arsenic on bacterial growth. For example, the growth of bacterial isolates from soil and water from the Hazaribagh tannery industrial area in Bangladesh decreased with increasing concentrations of As (27). Goswami et al. (28) modeled the effect of As (arsenic trioxide, $As_2O_3$) on the growth of *Aeromonas hydrophila* and observed no significant change in bacterial growth within the range of 0–30.32 µM As. Notably, we did not observe AUC bimodality under anaerobic conditions. This observation is consistent with our suspicion that bimodal growth may stem from differences in oxygenation across wells, given anaerobic cultures are not exposed to oxygen.

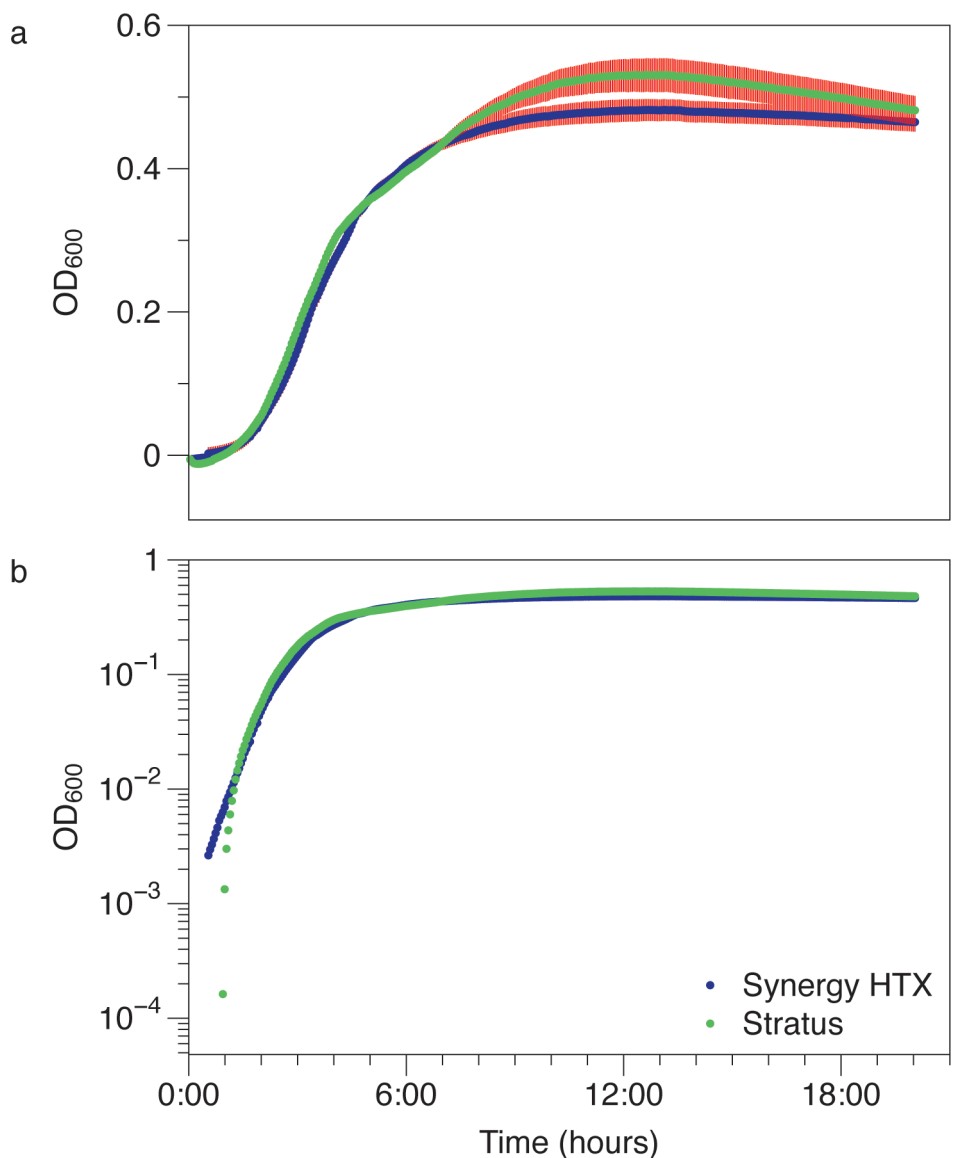

**FIG 3** The Stratus reader growth curves were similar to those obtained from the Synergy HTX. (a) Linear plot of average growth curves obtained from the two plate readers. Points are the mean $OD_{600}$ ± SEM ($n$ = 64 wells). When red error bars are absent, they are smaller than the symbol size. (b) Semilog plot of average growth curves obtained from the two plate readers. Points are the mean $OD_{600}$. No error is displayed.

## Considerations and difficulties

The Stratus microplate readers generally performed as expected in the anaerobic chamber. Yet, there were areas that we felt the Stratus under-performed, potentially limiting its utility for certain applications. First, the cultures were continuously shaken in anaerobic conditions during the duration of data collection. However, some anaerobes may be inhibited by agitation (29, 30). We did not test the Stratus in the absence of shaking or with periodic shaking. Importantly, the plate readers and shaking platforms are separate units, and there is no integrated method to systematically agitate the plate contents before obtaining time points. The PMS-1000i shaker is not programmable. Programmable plate readers could agitate the plate at regular intervals or immediately before a measurement, potentially improving the consistency of results.

Second, while conducting these experiments, we found the Stratus software and/or hardware to be inconsistent. We experienced connection issues and periodic changes or

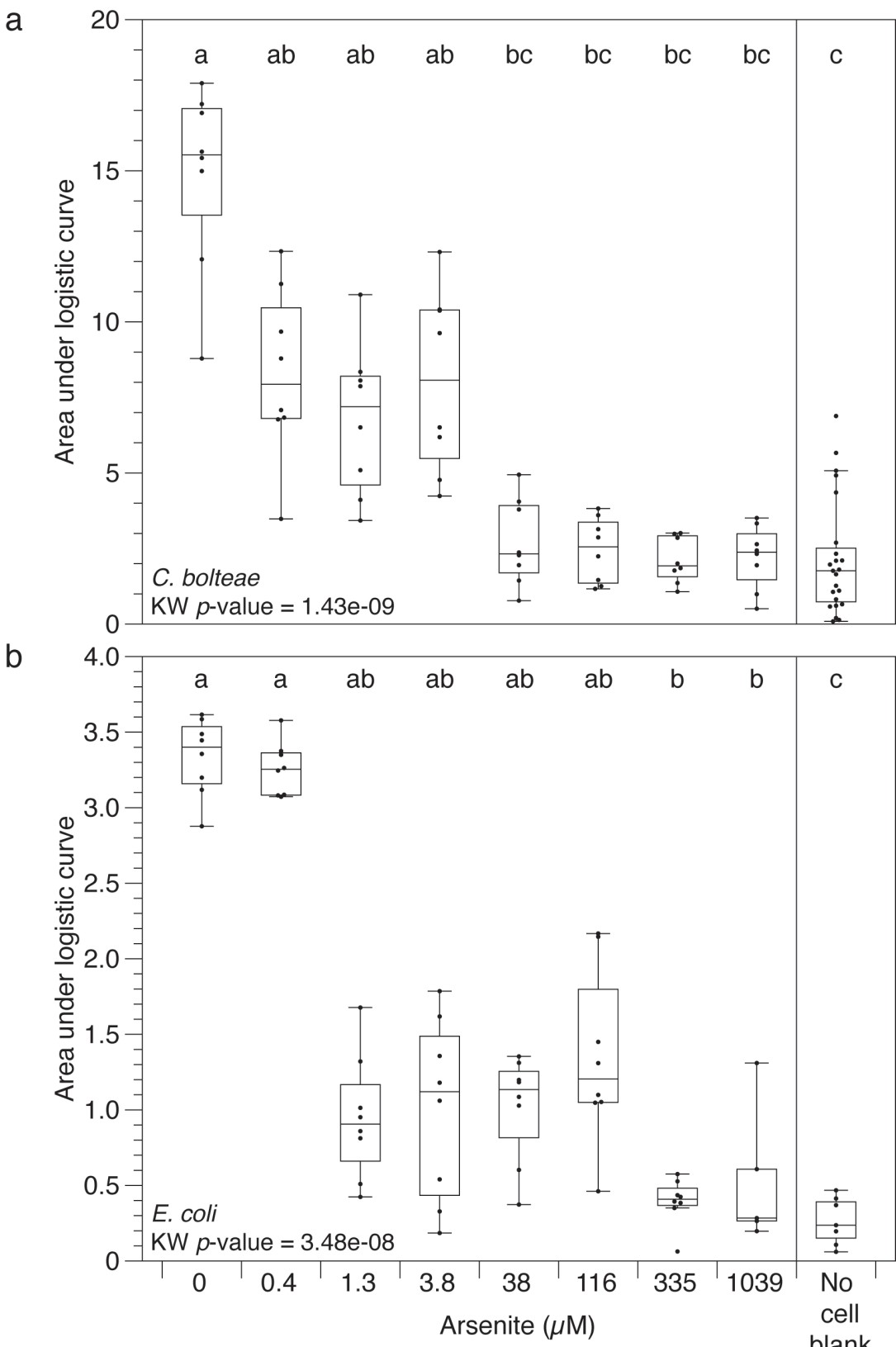

**FIG 4** Kinetic responses of (a) *C. bolteae* and (b) *E. coli* to arsenite while growing anaerobically. Points are the logistic AUC derived from growth curves obtained in microtiter plates incubated in the Stratus microplate reader. Box plots illustrate the interquartile range ± 1.5× interquartile range. The horizontal line in each box plot is the median. Outliers (>1.5× interquartile range) are shown as points. Box plots sharing letters are statistically indistinguishable by Dunn's test as defined by having a Bonferroni-corrected *P* value of ≥0.05. KW: Kruskal-Wallis.

bugs in the format of the data output. Some of these problems have been addressed with software updates from Cerillo. Occasionally the format of the data collected and stored on the .csv would change (usually coincident with a software update) without advance notification. This required us to recognize anomalies in our data, track down the change, and adjust our code accordingly. Third, handling the MicroSD card in the anaerobic chamber was not possible because of the small size of the MicroSD card and the reduced dexterity when using anaerobic chamber gloves. Thus, to retrieve the MicroSD card, the Stratus unit needed to be removed from the chamber, as there is no built-in wireless data transfer capability. Cerillo has since introduced a wireless accessory package allowing wireless access. However, this utility is only available with the purchase of their Software as a Service premium software. Finally, several practical constraints of the units as operated in anaerobic chambers should be considered. The Stratus is powered by a USB-C power adapter. The PMS-1000i microplate shaker requires a power cord. Thus, if the plate is shaken, two outlets are required per Stratus shaker combination. However, a solitary USB power hub could substitute for electrical outlets for the Stratus readers. Although the Stratus readers are designed to be stackable, we did not attempt to stack multiple Stratus readers on a single shaker and obtain growth readings, but stacking up to two Stratus readers on a single shaker may be possible.

## Conclusion

We present a workflow that relies on small-footprint microplate readers and an R script to analyze the resulting $OD_{600}$ data collected in anaerobic conditions. This approach provided results consistent with a name-brand microplate reader system. The small footprint saves substantial space in anaerobic chambers. We estimate that two Stratus-shaker systems fit within the same footprint as a single Synergy HTX. However, alternative shaker configurations may allow several Stratus instruments to be run simultaneously in the same footprint as the Synergy HTX. Although the small-footprint readers are more affordable than the larger systems, they are also more limited in what they can measure (OD at a single preset wavelength). Our pipeline facilitates the quick analysis of many samples inside an anaerobic chamber, increasing the speed of research and providing insight into anaerobe growth and metabolism. This method may be useful for the ever-expanding field of gut microbiome research where high-throughput cultivation will spur the discovery of novel metabolites and probiotics and facilitate dose-response studies of various drugs and toxins on the microbiome.

## METHODS AND EXPERIMENTAL DESIGN

### Bacterial cultures and media

Bacterial strains *Escherichia coli* DSM 28618 and *Clostridium bolteae* DSM 29485 were purchased from the German Collection of Microorganisms and Cell Cultures (DSMZ, Germany) and grown in the recommended media and incubation temperatures. *E. coli* was grown in LB (per liter, 10 g tryptone, 5 g yeast extract, 10 g sodium chloride, and adjusted to pH 7 with NaOH). *C. bolteae* was grown in Wilkins Chalgren Anaerobic Broth (CMO643 Oxoid Ltd). Media for anaerobic strains were prepared by boiling and cooling to room temperature while sparging under a constant 100% $N_2$ gas flow. All anaerobic media contained 0.5 mL $L^{-1}$ Na-resazurin solution (0.1% wt/vol) and L-cysteine-HCl (0.3 g $L^{-1}$). After autoclaving, media were equilibrated in an anaerobic chamber overnight. All anaerobic experiments were performed at 37°C inside a COY Model 2000, Forced Air Incubator for Vinyl Chambers, in a vinyl anaerobic chamber (Coy Laboratory Products, Inc., Grass Lake, MI, USA). All microplate-incubated cultures were conducted in 200 µL volumes in flat-bottomed polystyrene 96-well plates sealed with Breath-Easy Sealing Membranes and lids; the sealing membranes are thin films permeable to $O_2$, $CO_2$, and water vapor and UV transparent down to 300 nm. All growth curves were initiated with a 1:100 dilution of overnight cultures pre-incubated for ~18–20 h under aerobic or anaerobic conditions depending on the experiment.

## Aerobic and anaerobic OD data collection and experimental design

The plate in the Synergy HTX microplate reader was incubated inside the Synergy HTX at 37°C with continuous shaking at 180 rpm. The microplate incubated inside the Stratus was affixed to a PMS-1000i microplate shaker, shaking at 180 rpm inside a 37°C incubator. Both microplate readers were operating outside of an anaerobic chamber in aerobic conditions. The $OD_{600}$ was measured every 3 min overnight (~18–24 h).

For the arsenite dose experiments, individual treatments contained no (0 µM), or incremental micromolar doses of the cellular toxin As, as sodium arsenite ($NaAsO_2$): 0.4, 1.3, 3.8, 38.5, 115.5, 346, and 1,039 µM. We inoculated eight replicates per dose. Growth was monitored with an $OD_{600}$ reading taken every 3 min in anaerobic conditions at 37°C in a Stratus microplate reader (Cerillo, Charlottesville, VA, USA) shaking on a Microplate Shaker PMS-1000i (Grant Instruments, Shepreth, Cambridgeshire, UK) at 180 rpm. All experiments were conducted using kinetic mode with continuous shaking as the shaker is not programmable and cannot be stopped before an OD reading is taken.

## Growth curve modeling and kinetic parameter estimation

The $OD_{600}$ data were downloaded from the respective microplate readers and modeled with the Growthcurver R package (23). The Stratus output data were restructured using a custom R script to facilitate Growthcurver analysis.

## Statistical analysis

All statistical analyses, including Kruskal-Wallis, Mann-Whitney $U$, and Dunn's tests were conducted in R.

### ACKNOWLEDGMENTS

We thank Deanna Sanchez at the BIO5 Institute for the photograph of the Stratus Plate reader included in Fig. 1. We also thank Dr. Pawel Kiala for ordering the DSMZ cultures used in this study, and Coralee D'Agostino for help with the Growthcurver protocol.

Research reported in this publication was supported by the National Institute of Environmental Health Sciences of the National Institute of Health under award number P42 ES004940. The content is solely the responsibility of the authors and does not necessarily represent the official views of the National Institute of Health.

### AUTHOR AFFILIATIONS

[1]Department of Environmental Science, University of Arizona, Tucson, Arizona, USA
[2]School of Animal and Comparative Biomedical Science, University of Arizona, Tucson, Arizona, USA
[3]BIO5 Institute, University of Arizona, Tucson, Arizona, USA

### AUTHOR ORCIDs

Oona L. O. Snoeyenbos-West http://orcid.org/0000-0001-9146-7244
Christina R. Guerrero http://orcid.org/0009-0006-2441-0014
Paul Carini http://orcid.org/0000-0002-9653-7309

### FUNDING

| Funder | Grant(s) | Author(s) |
| --- | --- | --- |
| HHS | National Institutes of Health (NIH) | P42 ES004940 | Paul Carini |

### AUTHOR CONTRIBUTIONS

Oona L. O. Snoeyenbos-West, Conceptualization, Data curation, Formal analysis, Investigation, Methodology, Project administration, Supervision, Validation, Visualization,

Writing – original draft, Writing – review and editing | Christina R. Guerrero, Data curation, Formal analysis, Methodology, Software, Writing – review and editing | Makaela Valencia, Methodology, Writing – review and editing | Paul Carini, Conceptualization, Data curation, Formal analysis, Funding acquisition, Investigation, Methodology, Project administration, Resources, Supervision, Validation, Visualization, Writing – original draft, Writing – review and editing

## DATA AVAILABILITY

After processing the data with Growthcurver, we removed wells that did not model or for which the model was questionable as explained in Growthcurver literature. We also removed data from uninoculated blank wells that displayed growth due to carryover/contamination. All codes used are available at: https://github.com/cbaughan/CerilloWrangling

## ADDITIONAL FILES

The following material is available online.

### Supplemental Material

**Figures S1 and S2 (Spectrum03650-23-S0001.docx).** Growth curves of *E. coli* incubated in aerobic and anaerobic conditions.

### Open Peer Review

**PEER REVIEW HISTORY (review-history.pdf).** An accounting of the reviewer comments and feedback.

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
