## [Reviewer comments · Microbiology Spectrum]

Microbiology Spectrum

Cultivating efficiency: High-throughput growth analysis of anaerobic bacteria in compact microplate readers.

Oona Snoeyenbos-West, Christina Guerrero, Makaela Valencia, and Paul Carini

Corresponding Author(s): Oona Snoeyenbos-West, The University of Arizona

Review Timeline:

Submission Date:	October 12, 2023
Editorial Decision:	January 31, 2024
Revision Received:	February 27, 2024
Accepted:	February 29, 2024

Editor: Noha Youssef

Reviewer(s): The reviewers have opted to remain anonymous.

Transaction Report:

DOI: <https://doi.org/10.1128/spectrum.03650-23>

Re: Spectrum03650-23 (Cultivating efficiency: High-throughput growth analysis of anaerobic bacteria in compact microplate readers.)

Dear Dr. Oona Lesley Octavia Snoeyenbos-West:

Thank you for the privilege of reviewing your work. Below you will find my comments, instructions from the Spectrum editorial office, and the reviewer comments.

I apologize for the length of time it took to get reviews to you. Please address the reviewer's minor comments.

Revision Guidelines

Sincerely,
Noha Youssef
Editor
Microbiology Spectrum

Reviewer #1 (Comments for the Author):

The manuscript describes an exciting option for obtaining anaerobic growth curves that doesn't require much space in a glove bag. Overall, the manuscript was clearly written and should of interest to a large number of researchers who struggle to perform these assays with standard equipment. A few minor comments are listed below:

- The figure legends have become disassociated from the figures in the text.

- It would be helpful to see an overlay of either representative or average growth curves from the two plate readers. While the data is shown in the SI, the curves are small and hard to directly compare.
- Were there any differences in lag phase between the plate readers?
- Please update the wording of the actual setup. From the current language (and without one of the plate readers in hand), it's hard to understand if the plate was affixed to the shaker or if the Stratus is affixed to the shaker with the microplate inside the Stratus.
- Were the wells with larger AUC found in certain regions of the microplate? This would further suggest shaking differences and help researchers appropriately plan their experiments.
- Expanding upon the lack of bimodality under anaerobic conditions could help emphasize the potential cause for the two growth phenotypes.
- Please comment on whether the shaker is able to be programmed or if other shakers with programming capabilities are available.
- Was the signal to noise of the measurements during constant shaking higher than observed for plate readers that stop shaking prior to reading?

Reviewer #1 (Comments for the Author):

The manuscript describes an exciting option for obtaining anaerobic growth curves that doesn't require much space in a glove bag. Overall, the manuscript was clearly written and should of interest to a large number of researchers who struggle to perform these assays with standard equipment. A few minor comments are listed below:

We thank the reviewers for their positive feedback. Below, we have responded to each of the points in line in GREEN.

1.- The figure legends have become disassociated from the figures in the text.

We have moved all figures and captions to the end of the manuscript. See the new “Figures” and “Figure Captions” sections.

2.- It would be helpful to see an overlay of either representative or average growth curves from the two plate readers. While the data is shown in the SI, the curves are small and hard to directly compare.

We included a new Figure (Fig. 3) in the manuscript showing a plot of the average growth curves from both plate readers, including measures of error, to illustrate that curves generated by the Stratus reader closely resemble those produced by the Synergy HTX.

3.- Were there any differences in lag phase between the plate readers?

The new Fig. 3a shows there are negligible differences in lag phases between the two readers. We have noted in the manuscript that there is negligible difference in lag phase between the two plate readers.

4.- Please update the wording of the actual setup. From the current language (and without one of the plate readers in hand), it's hard to understand if the plate was affixed to the shaker or if the Stratus is affixed to the shaker with the microplate inside the Stratus.

The stratus was affixed to reader with the microplate inside the stratus. We have updated our wording regarding the setup of the plate readers for clarity.

5.- Were the wells with larger AUC found in certain regions of the microplate? This would further suggest shaking differences and help researchers appropriately plan their experiments.

We have addressed this question in the text of the manuscript noting that wells with larger AUC do not appear to be restricted to certain areas of the plate.

6.- Expanding upon the lack of bimodality under anaerobic conditions could help emphasize the potential cause for the two growth phenotypes.

We emphasize that we do not know what causes the bimodality. Yet, we have included additional commentary on the lack of bimodality under anaerobic conditions and related this to aerobic phenotypes being impacted by varying degrees of oxygenation due to shaking differences, whereas anaerobic growth was not impacted by minor variations in shaking.

7.- Please comment on whether the shaker is able to be programmed or if other shakers with programming capabilities are available.

We have included commentary on the shaker for the Stratus plate reader being not programmable. Several programmable shakers on the market may work. But these were not tested. For our purposes, the plate reader we chose was selected because it could be contained in the incubator we had in the anaerobic chamber. Other research groups may have other options available to them and any number of programmable shakers may work. We added a few words to this effect in the revision.

8.- Was the signal to noise of the measurements during constant shaking higher than observed for plate readers that stop shaking prior to reading?

Fig. 3a shows the error associated with reading replicate wells. The size of error bars for readings collected by the Stratus were larger than those collected by the Synergy HTX. However, we cannot be sure if this is solely due to the Synergy stopping before measurement. We are unable to test this in the absence of a system to stop the Synergy from shaking before measurement.

Re: Spectrum03650-23R1 (Cultivating efficiency: High-throughput growth analysis of anaerobic bacteria in compact microplate readers.)

Dear Dr. Oona Lesley Octavia Snoeyenbos-West:

Thanks for addressing the reviewers comments.

Your manuscript has been accepted, and I am forwarding it to the ASM production staff for publication. Your paper will first be checked to make sure all elements meet the technical requirements. ASM staff will contact you if anything needs to be revised before copyediting and production can begin. Otherwise, you will be notified when your proofs are ready to be viewed.

Sincerely,
Noha Youssef
Editor
Microbiology Spectrum